# A multi-scale microstructure to address the strength-ductility trade off in high strength steel for fusion reactors

Peng Gong [1,2] ✉, T.W.J. Kwok [3,4], Yiqiang Wang[5], Huw Dawson[5], Russell Goodall[1], David Dye [2] & W. Mark Rainforth [1] ✉

Fusion reactor materials for the first wall and blanket must have high strength, be radiation tolerant and be reduced activation (low post-use radioactivity), which has resulted in reduced activation ferritic/martensitic (RAFM) steels. The current steels suffer irradiation-induced hardening and embrittlement and are not adequate for planned commercial fusion reactors. Producing high strength, ductility and toughness is difficult, because inhibiting deformation to produce strength also reduces the amount of work hardening available, and thereby ductility. Here we solve this dichotomy to introduce a high strength and high ductility RAFM steel, produced by a modified thermomechanical process route. A unique multiscale microstructure is developed, comprising nanoscale and microscale ferrite, tempered martensite containing fine sub-grains and a high density of nanoscale precipitates. High strength is attributed to the fine grain and subgrain and a higher proportion of metal carbides, while the high ductility results from a high mobile dislocation density in the ferrite, subgrain formation in the tempered martensite, and the bimodal micro-structure, which improves ductility without impairing strength.

Nuclear fusion energy has long been regarded by many as a potential potent source of non-intermittent, low carbon electricity[1,2]. Fusion is attractive due to the abundance of fuel (hydrogen and its isotopes)[3] and short lifespan of the radioactive waste products[4]. However, the service conditions in fusion reactors are extreme, with components subjected to irradiation, neutron bombardment, exposure to helium and hydrogen, and very high temperatures[5–11]. In particular, within the plasma-facing fusion first wall and breeder blanket a significant effort is required to develop structural materials behind the plasma-facing surfaces that can survive such conditions (>600 °C) for realistic plant lifetimes; at least years for breeder blanket modules[1,12–15]. It is important that these materials can be manufactured at scale for future demonstration and commercial fusion power plants, such as the

European DEMO (EU DEMO) or the UK Spherical Tokamak for Energy Production (STEP) programmes[16–18], and multi-tonne conventional production routes are attractive compared to the need to establish new process routes and supply chains, which require significant investment and time.

Currently, some of the most promising materials for the breeder blanket are Reduced Activation Ferritic/Martensitic (RAFM) steels, due to their superior thermal conductivity, relatively low thermal expansion and resistance to radiation-induced swelling and helium embrittlement[12,19–24]. Despite international efforts to develop RAFM steels since the 1980s, and more recently in China, Russia, India and South Korea, the utilization of current RAFM steels is limited. There are some important aspects that will restrict the use of current RAFM

[1]Department of Materials Science and Engineering, University of Sheffield, Sir Robert Hadfield Building, Mappin Street, Sheffield S1 3JD, UK. [2]School of Materials, University of Manchester, Oxford Road, Manchester M13 9PL, UK. [3]Singapore Institute of Manufacturing Technology (SIMTech), Agency for Science Technology and Research, 5 Cleantech Loop, 636732 Singapore, Singapore. [4]Department of Materials, Royal School of Mines, Imperial College London, Prince Consort Road, London SW7 2BP, UK. [5]United Kingdom Atomic Energy Authority, Culham Science Centre, Abingdon OX14 3DB, UK. ✉e-mail: peng.gong@manchester.ac.uk; m.rainforth@sheffield.ac.uk

steels; for example, irradiation induces hardening and embrittlement at lower service temperatures (250–350 °C) and loss of creep strength and embrittlement at high operating temperatures (550–650 °C)[25–31]. To address this, developments seek to either achieve fully martensitic structures to avoid phase boundaries and abnormal growth of ferrite grains[32], or introduce an extremely high number density of nanoscale precipitates for strengthening at high temperature and to absorb irradiation defects, for example ODS-RAFM steel[33,34]. However, fully martensitic structures lead to reduced ductility, and irradiation induced effects limits the application temperature to 450–500 °C. It is also important to note the production of ODS steels is limited to small quantities and results in enhancing hardening performance at lower service temperatures.

Unlike automotive steels, which are designed to either resist deformation (anti-intrusion) or to deform and absorb large amounts of energy in a crash scenario, RAFM steels are not required, nor expected, to plastically deform in-service. Rather, the focus is to resist (micro-) cracking and damage, with better high temperature creep resistance. The high operating temperatures in the fusion reactor can lead to very large thermal stresses which may result in catastrophic material failure in the presence of stress concentrators, *e.g.* cracks, voids or other features on the phase boundaries. Therefore, it is expected that by improving the room temperature elongation to failure, it will be possible to extend the high temperature service life of RAFM steels and improve their resistance to irradiation-induced embrittlement. Therefore, excessive strain- or irradiation hardening is undesirable, while at the same time, ductility, toughening and the ability to resist cracking, e.g. at notches, is desired.

In a simple single-phase polycrystalline material, the onset of dislocation slip will occur in the grains with the highest Schmid factor, which results in load transfer to the surrounding grains and eventually through the yield transition to the propagation of deformation to every grain in the material. Subsequent work hardening can be relatively limited. This is problematic, as once the work hardening rate drops below the yield stress, the material can neck at any geometric imperfection. Therefore, as the yield stress is raised, tensile ductility and the ability to blunt a crack generally drops, which gives the well-known strength-ductility trade-off. This is the reason why a single process, such as work hardening, is not able to increase strength without a penalty to ductility. Thus, a range of strengthening mechanisms are required within a single material, often at different length scales, which operate harmoniously to simultaneously provide high strengthening and ductility[35–40].

Here, we extend this concept of a spectrum of deformation scales to RAFM steels. By designing a modified thermomechanical process route we have been able to produce 3 distinct, heterogeneous ferrite/martensite grain size populations and promoted a fine distribution of MC carbides allowing the combination of high strength and ductility. The ferrite phase is usually avoided due to the ease with which it coarsens, however, we show that ferrite with a non-uniform grain size can in fact be used to enhance the damage tolerance of the steel. The modified process route induces an extremely high dislocation density throughout the microstructure. During heat treatment the high dislocation density subsequently induces an extremely high number density of nanoscale precipitates, and importantly replaces a significant fraction of the $M_{23}C_6$ by cubic (Ti, V)C intragranular carbides, giving better high temperature stability. This approach to the RAFM steel microstructure can extend uniform elongation, without over-reliance on strain hardening for improvement of the ductility.

## Results
### Processing
A modified thermomechanical manufacturing process was developed, shown schematically in Fig. 1, to provide a multi-scale ferrite/martensite structure, expressly designed to give improved strength and ductility. Eurofer97 RAFM steel with the nominal composition of Fe-0.11C-9Cr-1.1W-0.2V-0.07Ta-0.4Mn-0.25Si-0.01Ti was used as the baseline. In addition, a steel with the same composition as Eurofer97, but with the addition of 0.25Si (wt%), was investigated, Supplementary Table 1. Si improves strength and ductility, accelerates strain induced ferrite formation, and is generally known to retard cementite formation on cooling of austenite in bainitic ferrite[34]. After reheating the slab to the soaking temperature and breakdown rolling in the austenitic temperature regime, rolling was performed in 3 stages. In the austenitic temperature regime during Stage 1 (1150–1100 °C) the steel is unable to fully recrystallise due to the relatively high alloy content. Instead, partial recrystallisation results in a highly deformed unrecrystallised austenite core ($\gamma_1$), decorated by fine recrystallised austenite grains ($\gamma_2$) on the grain boundaries[38,39].

The steel was then rolled in Stage 2 at 950–900 °C, just above the austenite-to-ferrite transformation, in order to bring about Deformation Induced Ferrite Transformation (DIFT)[41–43]. This nucleates DIFT ferrite grains ($\alpha_1$) on both the $\gamma_1$ and $\gamma_2$ grain boundaries. When this is quenched following rolling, a bimodal microstructure with the DIFT ferrite grains ($\alpha_1$) retained at room temperature and the austenite transforming to martensite ($\alpha'_1$). Even though the finer $\gamma_2$ grain size suppresses Ms[44], the finer necklace $\gamma_2$ grains can still transform to martensite. However, microstructural inspection suggests that some of the finer dynamically recrystallised austenite grains, $\gamma_2$, transformed to ferrite ($\alpha_2$), which implies some transformation took place during the initial stages of cooling.

In Stage 3 the steel was rapidly cooled by spray quenching to a warm intercritical ($\alpha + \gamma$) temperature, 850–800 °C, and then immediately rolled. All the large precursor $\gamma_1$ grains, small necklace $\gamma_2$ grains, and the DIFT ferrite grains ($\alpha_1$) are co-deformed. Recrystallisation of the austenite does not occur as the rolling temperature is below the recrystallisation stop temperature ($T_{NR}$) and so the austenite becomes elongated. Additional DIFT ferrite grains are also formed, $\alpha_1$, which have a different grain size to the $\alpha_1$ grains formed in Stage 2 rolling, yielding three grain size modalities. When the steel is then quenched after warm rolling, the large unrecrystallised $\gamma_1$ grains transformation to martensite $\alpha'_1$, while the DIFT ferrite grains are retained to room temperature.

Both Stage 2 and Stage 3 variants were then normalised at 980 °C for 1 h followed by quenching to room temperature. The $\alpha_1$, $\alpha_2$ and $\alpha'_1$ transforms to austenite; the Stage 2 steel with two grain size modalities, while the Stage 3 has three grain size modalities, namely $\gamma_3$, $\gamma_4$ and $\gamma_5$ in increasing size. Grain growth is retarded by the precipitation of $M_{23}C_6$ (M = Cr,Fe) carbides on the grain boundaries of $\gamma_3$, $\gamma_4$ and $\gamma_5$. Each austenite grain also inherits its previous composition as the normalisation time of 1 h is insufficient for diffusion of larger substitutional species such as Cr and Mn. On quenching to room temperature, the larger $\gamma_5$ grains transform to martensite ($\alpha'_2$). Microstructural inspection indicates that the smaller $\gamma_3$ and $\gamma_4$ grains transform to ferrite ($\alpha_3$ and $\alpha_4$ respectively) on cooling. This is shown in Supplementary Fig. 1. The martensite transformation of $\gamma_5 \rightarrow \alpha'_2$ injected a large density of mobile dislocations into the surrounding $\alpha_4$ grains, similar to that observed in DP steels[45]. This high dislocation density was important as it provided nucleation sites for the MC to precipitate in the subsequent ageing, which was unique to the Stage 3 steel.

Finally, both steels were aged at 750 °C for 1.5 h, below the A1 temperature where the austenite-to-ferrite transformation is complete. This ageing heat treatment has the effect on the microstructures of (i) tempering the martensite ($\alpha_T'$) (i.e. C diffusion but not the movement of substitutional species), (ii) allowing the dislocations within the martensite laths to rearrange themselves into subgrains and (iii) nanoscale precipitate formation.

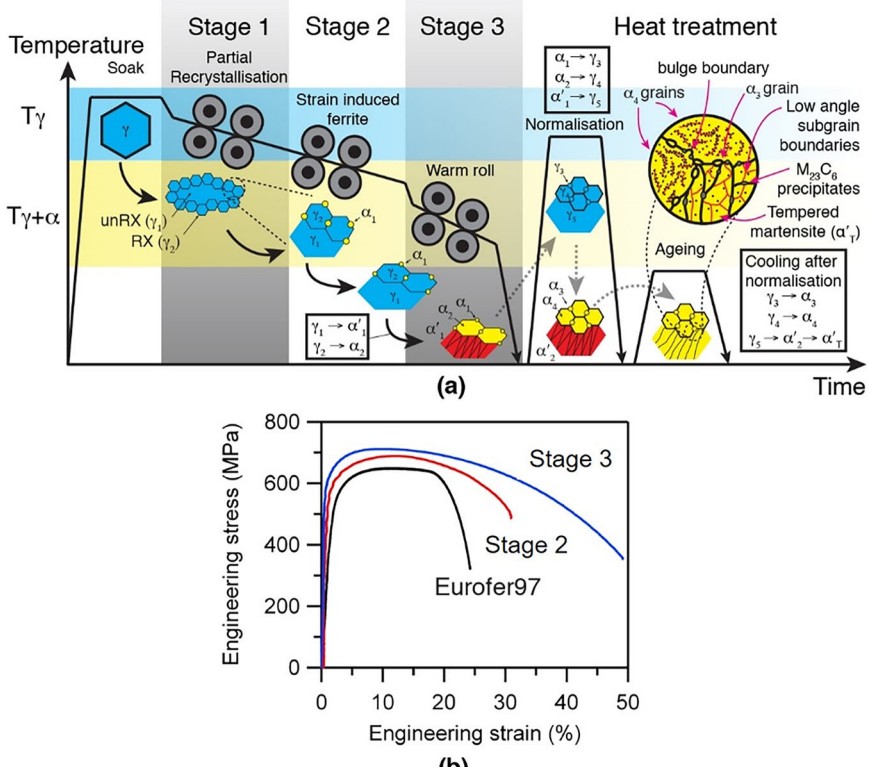

**Fig. 1 | Processing and resultant tensile properties of the alloys. a** Schematic of the modified thermomechanical manufacturing process and microstructural evolution. Note the bimodal microstructure is obtained without the warm rolling step but also with the same heat treatment. Blue shading indicates rolling in the fully austenitic region; yellow shading indicates rolling in the austenite + ferrite region. $\alpha_i$ (ferrite) and $\gamma_i$ (austenite) refer to the point at which each phase morphology was formed (see text). RX indicates recrystallised, while unRX indicates unrecrystallised. **b** Tensile behaviour of the reference material Eurofer97 and our Stage 2 and Stage 3 RAFM steels (graphs are a representative example taken from 3 repeat tests).

## Microstructure

Figure 2 shows the microstructures of the steel obtained using both the Stage 2 (Fig. 2a–d) and Stage 3 (Fig. 2e–l) processing routes. Further images showing the difference in microstructure between Stage 2 and Stage 3 are given in Supplementary Fig. 2. In Fig. 2a, e, the short chains of $\alpha_4$ necklace grains are observed using Electron Backscattered Diffraction (EBSD). These can be distinguished from the tempered martensite grains as the latter have an abundance of Low Angle Grain Boundaries (LAGBs, red lines) within each grain (high angle grain boundaries are in black). The grain size distributions from the EBSD measurements (above 5 μm, with a significant fraction below this, but only resolvable in the TEM) are given in Supplementary Fig. 3. The Stage 3 structure had a higher proportion of fine grains (<10 μm) than the Stage 2. The ferrite fractions were measured as ~43% for the Stage 2 and ~33% for the Stage 3.

As a comparison, the microstructure obtained after Stage 2 processing is shown in Fig. 2a–d. Without Stage 3, the necklace $\alpha_4$ and tempered martensite $\alpha_T'$ grains are slightly larger than the microstructure after Stage 3 processing (Fig. 2e–h). Figure 2c, g shows the substructure of the $\alpha_T'$ martensite laths for both stages, composed of subgrains, which also possess a high residual dislocation density. In Fig. 2f, a bowed boundary is observed, where the boundary unpinned itself from carbides, due to larger interparticle spacing, but is still being retarded/pinned by adjacent carbide precipitates, only allowing localised movement of the boundary. Furthermore, Fig. 2b, f reveals a number of $(Fe,Cr)_{23}C_6$ carbides, decorating both the ferrite and tempered martensite grain boundaries in both Stage 2 and Stage 3 microstructures. A bright field STEM micrograph and corresponding STEM-EDS images are shown in Fig. 2i–l; $(Fe,Cr)_{23}C_6$ carbides are mostly confined to the tempered martensite.

The dislocation structure and dislocation density in the $\alpha_4$ grains is significantly different between the Stage 2 and Stage 3 processing routes, Fig. 2d, h. Measured dislocation densities are given in Table 1. In the Stage 3 steel $\alpha_4$ grains, the high dislocation density present after cooling from normalisation rearrange to form Low Energy Dislocation Structures (LEDS)[46,47]. These dislocations also facilitate pipe diffusion, forming high number density of nanoscale MC carbides (~15 nm) (nominally, $(Ti, V)C$, see Supplementary Figs. 4 and 5) on and in the immediate vicinity of these dislocations, effectively pinning the LEDS in place after Stage 3 processing (Fig. 2h). This is compared to Stage 2 rolling where only a random distribution of dislocations pinned by sparse nanoscale carbides in Fig. 2d is observed. During ageing, there is also pressure for the $\alpha_4$ grains to coarsen in the Stage 3 processed steels. The $\alpha_4$ grain boundaries are highly decorated with either carbides or significantly finer $\alpha_3$ grains. These pinning particles effectively prevent the migration of grain boundaries resulting in the formation of curved interfaces (Fig. 2f).

Small-Angle Neutron Scattering (SANS) experiments have been undertaken for the measurement of the precipitation density, Fig. 3. Figure 3a shows one-dimensional plots of nuclear scattering intensity versus scattering vector on the Stage 3 RAFM steel developed here, Eurofer97 and pure iron, respectively. Taking the ratio of magnetic to nuclear scattering $(R(q))$, we can determine that the Stage 3 RAFM steel has much lower $R(q)$ (a value of ~1) than the baseline Eurofer97 (a value of ~2), indicating a lower fraction of $(Fe,Cr)_{23}C_6$ type precipitates larger than 150 nm (calculations in Supplementary Table 3[48–54]). This is significant, as these grain boundary carbides are believed to be deleterious to creep performance[55]. The population of ~15 nm diameter cubic $(Ti, V)C$ intragranular carbides was found by SANS to make up 0.16% volume fraction in the Stage 3 steel, shown in Fig. 3 and Figs. S2 and S3,

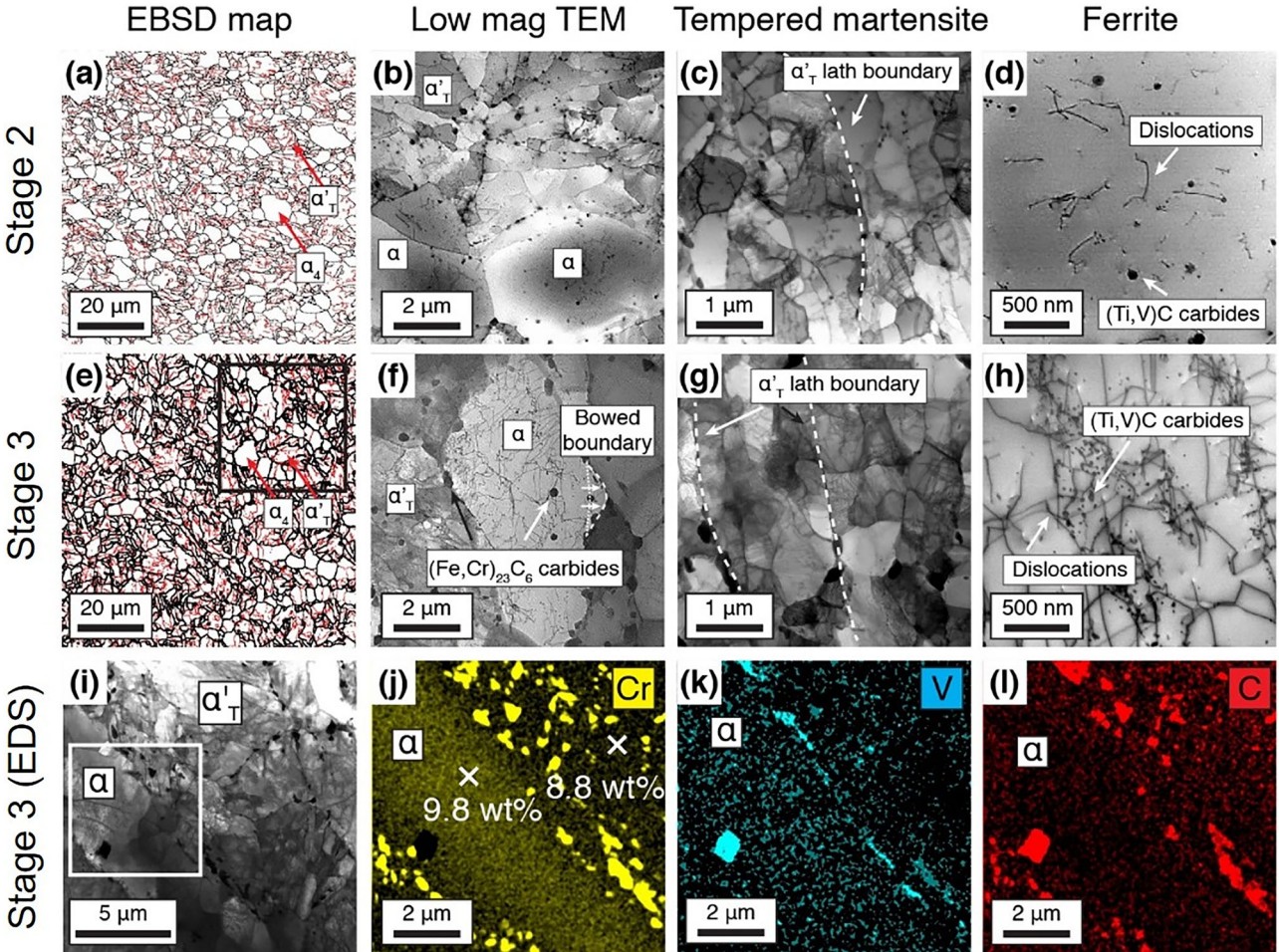

**Fig. 2 | Microstructure as a function of process history. a–d** Microstructures of the comparative Stage 2 RAFM steel and **e–h** Stage 3 processing route RAFM steel. **a** Electron backscatter diffraction (EBSD) grain boundary map of the Stage 2 RAFM steel. Bright field (BF)-TEM micrographs of **b** the general microstructure, **c** tempered martensite grains and **d** showing the relatively low dislocation and precipitate density within an $\alpha_4$ grain. **e** EBSD grain boundary map of the Stage 3 RAFM steel. Black lines indicate high angle grain boundaries (HAGBs) (>5˚) and red lines indicate low angle grain boundaries (LAGBs) (<5˚). BF-TEM microstructures of **f** an $\alpha_4$ necklace grain, **g** tempered martensite grains showing a subgrain structure and **h** dislocations pinned by fine precipitates within an $\alpha_4$ grain. **i–l** BF-TEM and EDS maps of a ferrite + tempered martensite region in the Stage 3 RAFM steel, showing the $Cr_{23}C_6$ and (Ti, V)C carbide locations. In the V map, finer scale (V, Ti)C are also visible (see Supplementary Fig. 4 and Supplementary Fig. 5).

**Table 1 | Measured and predicted yield strength with the contributions of the lattice friction stress $\Delta\sigma_O$, grain boundary strengthening, $\Delta\sigma_{GB}$, dislocation density ($\Delta\sigma_\rho$), precipitate strengthening ($\Delta\sigma_P$)**

| Material | $\Delta\sigma_O$ (MPa) | $\Delta\sigma_{GB}$ (MPa) | $\Delta\sigma_\rho$ (MPa) | $\Delta\sigma_P$ (MPa) | Predicted $\Delta\sigma_{total}$ (MPa) | Measured $\Delta\sigma_{total}$ (MPa) |
|---|---|---|---|---|---|---|
| Stage 2 | 48 | 188.9 | 33.1 | 139 | 409 | 398 |
| Stage 3 | 48 | 196.9 | 72.3 | 313 | 630 | 587 |

while the fraction was nearly zero in Eurofer97 steel. These nanoscale carbides would be expected to improve strength, without inhibiting ductility, while potentially providing higher tolerance to neutron damage.

## Tensile properties

The tensile properties of both microstructures are shown in Fig. 1b, obtained using full-size ASTM E8 sheet samples[56]. The full tensile data, including values of the uniform elongation, are given in Supplementary Table 2. Both Stage 2 and Stage 3 samples had a higher yield strength than the baseline Eurofer97 RAFM steel. Interestingly, the addition of the Stage 3 warm rolling gave higher yield strength, most likely owing to improved precipitation strengthening rather than the final dislocation density per se, as evidenced by the SANS bulk measurement results (Fig. 3). Moreover, what was striking was the significantly improved total and post-uniform elongation, which lies well outside the normal "banana" relationship for current RAFM and Dual Phase steels, Fig. 1c.

In order to investigate the origin of the impressive mechanical properties, the accumulation of damage during tensile testing of the Stage 3 samples, a series of interrupted tensile tests were conducted at engineering strains of 9% (true strain: 0.09), which is in the region of uniform elongation. In addition, samples were taken from the post uniform elongation region, namely at 16% (true strain: 0.145), 38% (true strain: 0.32) and at failure (49% (true strain 0.49)), Fig. 4, Supplementary Fig. 6a. These strains were selected on the basis of the work hardening behaviour, Supplementary Fig. 6a. The arrangement of dislocations into cells occurred at a strain of 9%, Fig. 4b–d, pinned by (Ti, V)C precipitates

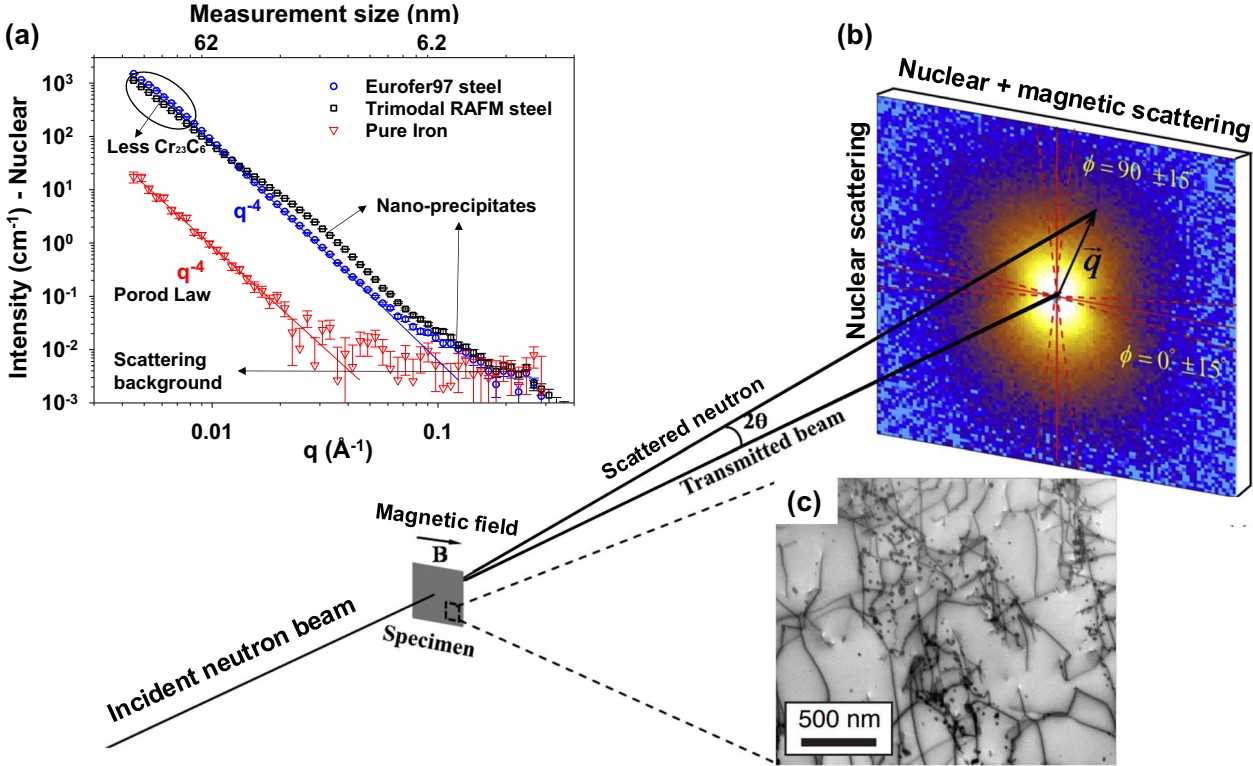

**Fig. 3 | SANS experimental set up and one-dimensional scattering intensities.**
Experimental configuration used in the current SANS measurements. **a** One-dimensional nuclear scattering intensities versus scattering vector obtained from the Eurofer97, the present Stage 3 RAFM steel, and pure iron (as a reference). Error bars show the standard error. (See Methods, section "Small angle neutron scattering", for the Porod Law which was discovered by Günther Porod, and describes the asymptote of the scattering intensity I(q) for large scattering wavenumbers q; q defined in Eq. (8)). The arrows show the direction the curve goes in for a reduction in $Cr_{23}C_6$ precipitates and nanoprecipitates such as MX. **b** An incident neutron beam transmitted through a bulk specimen containing nano-sized precipitates embedded in a ferritic matrix. **c** The resultant SANS two-dimensional pattern in the presence of a horizontal magnetic field. $2\theta$ is the scattering angle, $q$ is the scattering vector.

at the cell boundaries. By a strain of 16%, the dislocation density in cell walls increased substantially and these became elongated in the applied stress direction with increasing strain until failure (49% strain). The cell interiors have relatively low dislocation density, but in some there are intense slip bands (Supplementary Fig. 7). Such evidence of planar slip in ferrite has been reported in austenite-ferrite dual phase steels when grain rotation is inhibited[57,58]. In this case it is likely that the intense slip bands are due to the very fine ferrite grain size. With continued increase in strain to 38%, two planar slip systems are observed within the network cell structures, (110)[111] and (112)[111], Supplementary Fig. 7d. The lack of forest hardening therefore indicates that there may be strain softening in the ferrite. However, there may nevertheless be considerable post-uniform elongation as the ferrite phase remains soft and ductile up to fracture. Thus, planar slip and elongation of subgrains, leading to softening, combined with the formation of dislocation cell structures in the ferrite Fig. 4d, i, n, s, causing strengthening, significantly improves the mechanical properties to achieve extremely high post-uniform elongation deformation.

At a strain of 38% a new microstructural feature was observed, hitherto not reported. New fine scale (<100 nm), strain-free, ferrite grains appeared, first in the EBSD-Transmission Kikuchi Diffraction (TKD) map (Fig. 4k, i), with the number density increasing at a strain of 49% in BF-STEM as well as the TKD map (Fig. 4p, q). No such strain-free grains were present in the material prior to tensile testing. These strain-free grains, Supplementary Fig. 8, formed at the ferrite - tempered martensite

interfaces at the regions of highest kernel average misorientation, an indicator of GND density.

Figure 5 shows TKD maps from near the fracture surface (necked region). In Fig. 5a–c voids can be observed at both the ferrite/ferrite and ferrite/tempered martensite boundaries. Voids on grain boundaries are not conventionally associated with positive effects. In dual phase (DP) steels higher volume fraction of voids on the dual phase boundaries leads to shorter post-necking elongation. This occurs as those voids act as stress concentrations and can lead to dual phase boundary failure. However, it is interesting to note that quite a few of the voids remained small (~100 nm), despite the material experiencing post-instability deformation.

## Discussion
The effect of the microstructural features developed by this process on both strength and ductility is significant. An increase in yield strength is classically obtained through a reduction in grain size (and with a reduction in grain size distribution[59]). However, the decreased ductility associated with fine grain sizes is a major limitation. For example, ODS steels have high strength, but poor impact toughness, which is a problem for fusion applications, particularly with irradiation-induced hardening and embrittlement problems.

The yield strength of the material, $\Delta\sigma_{total}$ can be predicted from refs. 60,61:

$$\Delta\sigma_{total} = \Delta\sigma_0 + \Delta\sigma_{SS} + \Delta\sigma_{GB} + \Delta\sigma_\rho + \Delta\sigma_P \quad (1)$$

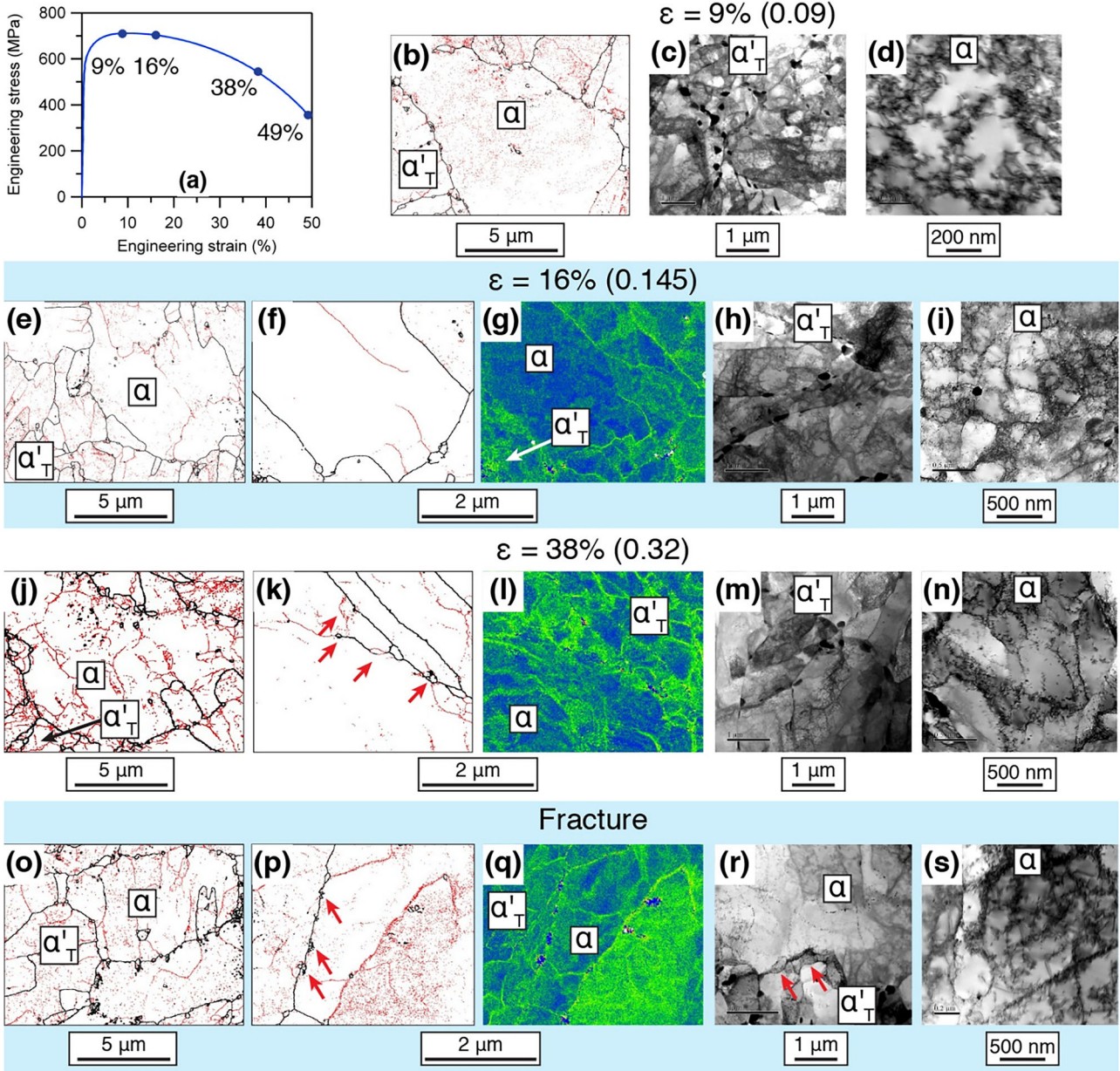

**Fig. 4 | Microstructures at different tensile strains. a** Engineering stress-strain curve of the Stage 3 RAFM steel. Markers indicate the strain to which interrupted tensile tests were conducted (curve taken as representative of 3 different repeat tests). **b**–**d** Sample strained to 9%: (**b**) grain boundary map, black lines indicate high angle grain boundaries (HAGBs) (>5˚) and red lines indicate low angle grain boundaries (LAGBs). **c** STEM-BF micrograph of martensite and (**d**) STEM-BF micrograph of precipitates and dislocations in the ferrite. **e**–**i** Sample strained to 16%: (**e**) grain boundary map; **k** Transmission Kikuchi Diffraction (TKD) boundary map and its corresponding Kernel Average Misorientation (KAM) in **l**; STEM-BF micrographs showing **h** martensite and **i** dislocation cell formation in ferrite. **j**–**n** Sample strained to 38%: (**j**) grain boundary map; (**k**) TKD boundary map with red arrows pointing to the nucleation of low orientation gradient ferrite grains along a ferrite/martensite boundary and corresponding **l** KAM map; STEM-BF micrographs showing **m** martensite and **n** elongated dislocation cell formation in ferrite. **o**–**s** Sample strained to 49%, i.e. failure. o grain boundary map; **p** TKD boundary map showing periodic formation of strain-free ferrite grains with corresponding (**q**) KAM map; STEM-BF micrographs of (**r**) newly formed strain-free grains at a ferrite/martensite boundary and **s** elongated subcell formation in ferrite.

is where $\Delta\sigma_0$ is the lattice friction stress, $\Delta\sigma_{SS}$ is the solid solute strengthening, $\Delta\sigma_{GB}$ is the grain boundary strengthening, $\Delta\sigma_{\rho}$ is the dislocation strengthening, and $\Delta\sigma_P$ is the precipitation strengthening.

The Peierls-Nabarro lattice friction stress $(\Delta\sigma_0)^{62}$ is given by:

$$\sigma_0 = (2G/1 - \upsilon)\exp(-2\pi w/b) \qquad (2)$$

where G is the shear modulus, $\upsilon$ is the Poisson's ratio, and $w$ is the dislocation width and b is the Burger's vector. Using a standard width of a dislocation[60,63] this gives a value of 48 MPa for the lattice friction stress.

The solid solution strengthening component can be calculated from ref. 64:

$$\tau_{ss} = \left(\sum_i k_{ss,i}^2 c_i\right)^{1/2} \qquad (3)$$

Where $k_{ss,i}^2$ is the strengthening coefficient of element $i$ in the matrix, and $c_i$ is the atomic fraction of element $i$ in the matrix. The value of $k_{ss,Cr}$ was taken as 82.7 MPa from reference[64], derived for an RAFM steel.

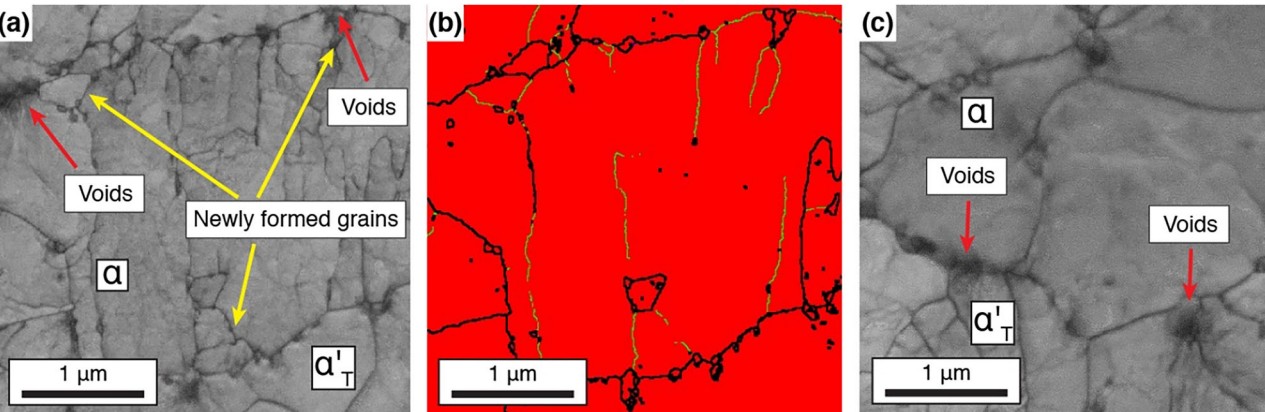

**Fig. 5 | Microstructure of the fractured Stage 3 sample.** Micrographs were obtained from a postmortem Stage 3 sample just below the fracture surface. **a** Transmission Kikuchi Diffraction (TKD) image quality map and corresponding **b** phase and boundary map showing several voids along a grain boundary. black lines indicate high angle grain boundaries (HAGBs) (>5˚) and red lines indicate low angle grain boundaries (LAGBs) red colouration is BCC ferrite. **c** TKD image quality map from another region showing similar voiding along grain boundaries.

The grain size contribution, ($\Delta\sigma_{GB}$) is based on the Hall-Petch relationship[65–67]:

$$\Delta\sigma_{GB} = K_y d_F^{-0.5} \tag{4}$$

where $K_y$ is a constant, taken as 0.55 MPa m$^{0.5}$ [60,68,69] and $d_F$ the average grain size measured in m.

The increment in the yield stress arising from the dislocation density ($\Delta\sigma_\rho$)[70]:

$$\Delta\sigma_\rho = \alpha M G b \sqrt{\rho} \tag{5}$$

is a material constant (0.33 used for steels[68]), M is the Taylor factor (taken as 2.75 assuming a random texture[69]), G is the shear modulus (80.3 GPa), b is the Burgers vector ($2.48 \times 10^{-10}$ m), and $\rho$ is the dislocation density. The estimates of dislocation density are given in Table 1.

The contribution of precipitation to the yield strength, $\Delta\sigma_P$ can be calculated using the Ashby-Orowan model[70,71]:

$$\Delta\sigma_P = 8995 * \frac{f_V^{1/2}}{d} \ln(2.417d) \tag{6}$$

Where $f_V$ is the volume fraction, and $d$ is the average diameter of the precipitates in nanometres.

Values of the measured grain size, dislocation density, precipitate volume fraction and precipitate size are given in Supplementary Table 3. These values were input into Eq. 1–6, with the resultant predictions of individual contributions and the total amount shown in Table 1.

The predicted yield strength is higher than the measured value for both Stage 2 and Stage 3 steels, with the prediction for the Stage 3 steel being about 8% higher than the measured value. This is most probably from errors in the experimental measurements of dislocation density and solid solution hardening, since these measurements were taken from TEM samples rather than bulk samples. Nevertheless, the values show the marked difference between Stage 2 and Stage 3 steels with the increase in dislocation density and the increase in volume fraction of fine MC precipitates induced by the Stage 3 processing resulting in a much higher yield strength.

The Stage 3 sample offered marginally better uniform elongation compared to the Stage 2 steel, which was better than the Eurofer97, Supplementary Table 2. However, the total elongation for the Stage 3 steel was a remarkable 49%, which far exceeds that found in current RAFM steels. The enhanced total ductility was believed to arise for three reasons: (a) the presence of ferrite in the Stage 2 and Stage 3 materials, compared to the Eurofer97 which comprised entirely of tempered martensite; (b) the bimodal grain structure and (c) the high density of mobile dislocations present in the Stage 3 steel.

Recent work has shown in several metal systems that a bimodal grain size can improve ductility without significantly impairing strength. The finer grains in the structure impart high strength while the larger grains exhibit high work hardening ability, leading to higher ductility[37,72–75]. For example, Patra et al.[73] investigated the role of a bimodal ferrite grain size on both the strength and ductility. They observed that the strength was primarily controlled by the finer grains. However, a higher strain accumulation was observed in the coarser grains, such that the coarser grains contributed to the higher ductility. The current work is consistent with this, with high strain accumulation being observed in the coarser ferrite grains, Fig. 4q, and evidence of the high dislocation density in Stage 3 samples being mobile, even though many were pinned by fine MC carbides.

Extensive deformation occurred after the onset of necking (i.e. after the limit of uniform deformation at 9% engineering strain) in the Stage 3 steel, which was not present in the Stage 1 or 2 steels. In conventional DP steels, voids form at stress concentrations at the martensite/ferrite interface, which ultimately limit ductility. In contrast in the present case, fine voids were observed at these interfaces, but these appeared stable and were not ductility limiting. Calcagnotto et al.[72] looked at the effect of ferrite and martensite grain size in DP steels on the deformation mechanisms and strain localisation and voiding. As the grain size was reduced, enhanced martensite plasticity and better interface cohesion was observed. In the finest grain size material, planar slip was observed in the ferrite, as was observed here (Fig. 4), which Tomota[76] considers to be a result of the restricted operation of plastic relaxation of strain incompatibility. Moreover, the martensite in the present steel is tempered, unlike DP steels, which will have improved co-deformation of two phases. This led to reduced voiding during tensile straining and an enhancement of ductility in the DP steel. The current observations of reduced and stable voiding in the Stage 3 material appear to be entirely consistent with the findings of Calcagnotto et al.[72] which is a clear contributing factor to the high tensile ductility observed for the Stage 3 processed material.

The key features in the microstructure that contributed to such high ductility include the high mobile dislocation density within the ferrite, which, with the absence of forest hardening, indicates that there may be strain softening in the ferrite. This allows considerable post-uniform elongation as the ferrite remains soft and ductile up to fracture.

**Table 2 | The rolling parameters including the strain rate, passes and reduction of per pass**

| Material | strain rate | passes | reduction of each pass |
|---|---|---|---|
| Stage 1 | $3s^{-1}$ | 2 | 3.5mm |
| Stage 2 | $3s^{-1}$ | 2 | 1.5mm |
| Stage 3 | $3s^{-1}$ | 2 | 0.75mm |

In addition, the higher proportion of fine MC carbides improve strength without greatly inhibiting ductility, and the nanoscale subgrains in martensite and nanoscale ferrite grains also both benefit the strength. The very fine new strain free nanograins were formed during tensile deformation are also clearly a reflection of stable deformation to high strain. Gholizadeh et al.[77] observed similar features in an examination of the high strain deformation behaviour of an ultra-low carbon IF steel. They observed the formation of "ultrafine grains" with dislocation free interiors bounded by high angle grain boundaries, which are similar to those grains observed here. They argued that these formed by the subdivision mechanism (where deformation-induced GNDs splitting the grains into finer domains that have different slip patterns), as detailed by Hughes and Hansen and others[78–81]. Misorientation increases across the boundary as the strain increases as a result of lattice rotations. A similar observation of dislocation free fine grains forming during deformation has been made by Wang et al.[82] for the tensile deformation of a X80 line-pipe steel. They found regions of low dislocation density were bounded by regions of high dislocation density (high GND) arguing that the high dislocation density bands exerted a strong back stress that led to the dislocation free regions. In both these cases this behaviour was associated with high ductility.

In summary, a modified thermomechanical rolling process has been applied to an RAFM steel composition which produces a completely different microstructure to that conventionally seen in RAFM steel after heat treatment. This microstructure comprises micron-sized ferrite, ferrite with a size in the nanoscale range, and tempered martensite containing subgrain structures, combining fine (Ti, V)C precipitates formed on the high density of dislocations, further pinning the structure and adding to the strength. The newly designed RAFM steels achieved similar ductility to highly ductile interstitial free (IF) steels, but with substantially higher strength, giving an outstanding combination of strength and ductility. This approach provides an RAFM steel with both the desired high temperature strength, sufficiently low impact transition temperature and potentially high tolerance radiation damage.

## Methods

### Materials
The composition of the new RAFM steel investigated in this study is listed in Supplementary Table 1. The steel was produced as 20 mm × 20 mm × 200 mm in an arc melting furnace with an argon atmosphere. The ingots were then homogenized at 1250 °C for 1 h followed by quenching in oil to room temperature. The steels were then hot rolled at 1150–1100 °C down to a thickness of 12 mm for the first pass, following the second pass at 950–900 °C with reducing thickness to 8 mm. The Stage 2 samples were quenched at this stage. Stage 3 samples were then given a third pass operated at 850–800 °C down to the final thickness of 4 mm. The rolling parameters, such as strain rate, number of passes and reduction per pass are given in Table 2. Samples were austenitised at 980 °C for 1 h followed by water quenched and ageing at 750 °C for 1.5 h followed by air-cooling. A schematic diagram showing the overall process route is given in Fig. 1. All heat treatments were undertaken in an Ar atmosphere in order to prevent oxidation and decarburization. The Eurofer97 steel used in the current study was provided by Karlsruhe Institute of Technology (KIT) in the form of a 6 mm thick plate, with a chemical composition summarized in Supplementary Table 1.

### Tensile testing
Tensile testing was also conducted for each processing condition at a constant strain rate of $10^{-4}s^{-1}$ using the sample geometry as shown in Supplementary Fig. 9 at room temperature using a Zwick (BTC T1-FR020 TN A50) universal testing machine. Tensile tests were conducted three times for each condition.

### Microstructure analysis
Microstructural observations were performed by electron backscatter diffraction (EBSD), and transmission electron microscopy (TEM). Specimens for EBSD observation were prepared by standard metallographic methods, which included grinding from P400 to P1200 and polishing up to 0.04 µm colloidal silica particles. The EBSD analyses were undertaken on a FEI-Nova 600 and JEOL JSM 7900 operating at 20 kV with a beam step size of 0.15–0.1 µm. The HKL Channel5 system (Oxford Instruments, Oxfordshire, United Kingdom) was used for data acquisition and analysis. The microstructural constituents can be distinguished by EBSD on the basis of following parameters: (1) the band contrast (BC) map depends on the distortions of the crystal lattice; (2) Kernel Average Misorientation map (KAM) by analysing the average misorientation angle of a give point with all its neighbours. 6 Kikuchi lines and 1.5 angular tolerances were employed to index the Kikuchi patterns to prevent misidentification.

Thin specimens for TEM were cut from the heat-treated blanks, and ground to about 60 µm using conventional techniques. The foils for TEM studies were prepared by standard electropolishing. Electropolishing was performed in a Tenupol Model twin-jet electropolishing unit, using a solution of 50 ml perchloric acid, 600 ml methanol and 350 ml butyl alcohol at a temperature of −40 °C. TEM studies were conducted in a JEOL-F200 microscope operating at an accelerating voltage of 200 kV, equipped with a JEOL Dual-EDS system. The microstructural analyses were performance in STEM mode in order to reduce diffraction effects on the dislocation distribution. The dislocation density was measured by the intersection counting technique. Lines of a known total length were drawn on the image, and the number of intersections with the dislocations was counted. The dislocation density, ρ, was then calculated using the formula:

$$\rho = 2nlt \qquad (7)$$

Where $n$ is the number of intersections with the dislocations, $l$ is the length of the grid used, and $t$ is the thickness of the foil. The thickness of the foil was measured by electron energy loss spectroscopy (EELS) using the low loss peaks.

### Small-angle neutron scattering
Small-Angle neutron scattering (SANS) experiments were performed on the Sans2d and ZOOM beamlines at the ISIS Pulsed Neutron Source UK[70]. A magnetic field of ~1.6 T was applied to saturate the ferritic matrix which allows the separation of the magnetic and nuclear scattering, as shown in Fig. 3b. Specimens with dimension of ~10 mm × ~10 mm × ~1 mm were prepared with a final 1200 grit SiC grinding. A sample-to-detector distance of 4 m was used to provide scattering vector, $q$, covering a range of 0.004 to 0.3 $Å^{-1}$ as follows:

$$q = 4\pi \sin\theta / \lambda \qquad (8)$$

where $2\theta$ is the scattering angle and $\lambda$ is the neutron wavelength. The neutron beam size was 8 mm in diameter and the measurement time for each sample was 1.5 hrs. To avoid collecting the scattering signal from multi-Bragg diffraction, only neutrons with wavelengths, $\lambda$, from 4.7–16.5 Å were selected for data analysis[71–74]. One-dimensional nuclear and "nuclear + magnetic" scattering intensity plots, $I$ (intensity) versus $q$ were obtained by partial azimuthal averaging in sectors around the horizontal and vertical axes of the transmitted beam respectively using the software Mantidplot[75], as shown in Fig. 3a.

**Table 3 | Values of nuclear ($\rho_{nuc}$) and magnetic ($\rho_{mag}$) scattering length density, nuclear ($\Delta\rho_{nuc}$) and magnetic ($\Delta\rho_{ma}$) contrast factor of phases potentially relevant to this study**

| Phase | $\rho_{nuc}$ $(10^{14}m^{-2})$ | $\rho_{mag}$ $(10^{14}m^{-2})$ | $\Delta\rho_{nuc}$ $(10^{14}m^{-2})$ | $\Delta\rho_{mag}$ $(10^{14}m^{-2})$ | $R(q) = (\Delta\rho_{mag}/\Delta\rho_{nuc})^2$ |
|---|---|---|---|---|---|
| Fe | 7.93 | 5 | / | / | / |
| $Cr_{23}C_6$ | 4.15 | 0 | 3.78 | 5 | 1.75 |
| VC | 3.43 | 0 | 4.5 | 5 | 1.23 |
| TiC | 1.63 | 0 | 6.3 | 5 | 1.55 |
| TiN | 3.04 | 0 | 4.89 | 5 | 1.04 |

Detailed calculation can be found in [6]. Precipitate phases which are paramagnetic generally considered to have a zero magnetic scattering length density.

Supplementary Fig. 10a, b show one-dimensional plots of magnetic (blue) and nuclear (black) scattering intensity versus scattering vector for the Stage 3 RAFM steel and the EUROFER97 steel, respectively. In the Eurofer97 steel, both the magnetic and nuclear signals follow a $q^{-4}$ variation (known as the Porod Law) over the entire $q$-range until the background level is reached at $q > 0.1$ Å$^{-1}$. This indicates that the primary contribution to scattering is from large particles or features with size typically $>2\pi/q_{min}$ (>150 nm). These will typically be metallic carbides (i.e., $(Fe,Cr)_{23}C_6$). The magnetic scattering contribution of $(Fe,Cr)_{23}C_6$ in Eurofer97 steel is larger than the nuclear scattering contribution because of the larger magnetic contrast factor compared to the nuclear contrast factor (Table 3). The current SANS measurement shows that the number density and volume percent of MX type nanoscale precipitates formed in Eurofer97 steel is extremely low.

In contrast, the SANS results obtained from the Stage 3 RAFM steel indicate that both magnetic and nuclear scattering intensities deviate from a $q^{-4}$ Porod-type behaviour at $q\sim 0.007$ Å$^{-1}$. This deviation arises from the formation of nanoscale ($d \leq 60$ nm) MX precipitates.

To determine the average chemical composition of the precipitate in both Eurofer97 and the Stage 3 RAFM steel samples, the corresponding ratio of magnetic to nuclear scattering, R(q), was calculated and is shown in the left-hand axis of Fig. 3 as red and square circles. The average value of R(q) exhibited a shift from ~2 (in Eurofer97) to ~1 (in the Stage 3 RAFM steel), indicating the formation of a new type of precipitate with a much lower R(q) value. Comparison with theoretical calculations (Table 3) suggest that R(q) = ~1 represents the formation of VC/TiN precipitates in the Stage 3 RAFM steel.

To further analyses the precipitate size and volume percent ($f_v$) in the stage 3 RAFM steel, the Porod region due to the large carbides and the incoherent background in the nuclear SANS signal were subtracted and plotted in Supplementary Fig. 11a. This was done to obtain a meaningful scattering signal only from nano-sized precipitates. The Kratky plot, $Iq^2$ versus $q$ is then plotted in Supplementary Fig. 11b. In the Kratky plot, a characteristic particle size, $R_{max}$, (the "pseudo-Guinier radius"), can be determined from the maximum $q_{max}$ in the plot[72] as follows:

$$R_{max} = \sqrt{3}/q_{max} = 9.6\text{nm} \tag{9}$$

$$q_{max} = 0.179\text{nm}^{-1} \tag{10}$$

The volume percent of the precipitate was calculated from nuclear SANS sign using equation:

$$Q = \int_0^\infty I(q)q^2 dq = 2\pi\left(\rho_\rho - \rho_m\right)^2 f_v(1 - f_v) \tag{11}$$

where $\rho_p$ and $\rho_m$ are the nuclear scattering length densities of precipitate and matrix respectively. The $q$-range for integration was extrapolated to $10^{-5}$ Å$^{-1}$ for low $q$ and 10 Å$^{-1}$ for high $q$ with the Guinier

equation and the Porod law, respectively, as recommended in ref. 72 Thus $f_v = 0.0016$ was calculated using the SasView software[76].

## Data availability
The tensile and SANS data generated in this study are provided in Figshare[83].

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

## Acknowledgements
This work is supported by the EPSRC grant EP/X030652/1, Royal Society Grant RG\R2\232517 and SUSTAIN Research Hub Early Career Research (application number ECRC1 014 Gong). The authors wish to acknowledge the Engineering and Physical Sciences Research Council (grant number EP/S018107/1) as a part of 'SUSTAIN Manufacturing Hub', and Henry Royce Institute for Advanced Materials, funded through EPSRC grants EP/R00661X/1 and EP/P02470X/1 for access to the JEOL JEM-F200 and JEOL JEM 7900 F. The authors also wish to acknowledge the DARE project (grant number EP/L025213/1) and Professor Cameron Pleydell-Pearce from Swansea University for their support in hot rolling. The authors also acknowledge the Science and Technology Facilities Council (STFC) for granting access to neutron beamtime at ISIS, ZOOM and Sans2d facilities. The authors also wish to thank the Karlsruhe Institute of Technology for supply of the Eurofer97 for the SANS measurement. This work has been partially funded by the EPSRC Energy Programme (grant number EP/W006839/1), supporting the time of Yiqiang Wang and Huw Dawson.

## Author contributions
P.G. designed the study, performed most of the experimental work, and wrote the manuscript. T.W.J.K. Kwok undertook melting and rolling of steels, figure construction and discussed the data. R.G. discussed the data and contributed to writing the manuscript. Y.W. undertook the SANS work, including the SANS experiments, data analysis, figure creation, and drafting the corresponding SANS sections. H.D. undertook heat treatment. D.D. and W.M.R. undertook scientific interpretation and wrote the manuscript.

## Competing interests
The authors declare no competing interests.
