## [Transparent Peer Review file · Nature Communications]

A multi-scale microstructure to address the strength-ductility trade off in high strength steel for fusion reactors

Corresponding Author: Professor William Rainforth

Version 0:

Reviewer comments:

Reviewer #1

(Remarks to the Author)

The novelty of this work consists on proposing a novel thermomechanical processing for RAFM steels that enhances simultaneously tensile strength and ductility by a considerable refining of the final microstructure formed. The key-factor for such behavior is the combination of warm-rolling stage where DIFT mechanism is activated, and a subsequent normalization treatment to generate a final ultrafine martensite grain microstructure.

This novel processing, which has been studied profusely for steel processing in automotive sector applied steels, has not been reported previously for RAFM steels. Therefore, this paper reports a work of significance for the field.

Regarding literature review of the state-of-the-art, the references indicated are fairly modern with well selected number of seminal papers on the matter. I miss some of earlier works on this subject such as:

N. Baluc et al, Journal of Nuclear Materials, 367-370 (2007) 33-41

H. Tamigawa et al, Fusion Engineering & Design, 83 (2008) 1471-1478

The work reported in this paper support the conclusion raised and is methodology sound, providing enough details to reproduce the tests.

I could not find any flaws in the paper. My only concern is all the tests performed are focused on improving strength-ductility compromise, but not dynamic mechanical testing were performed in order to check if this claimed improvement is really achieved. No creep tests on irradiated samples have been performed.

Having all the above mentioned reasons in mind, I recommend the acceptance of the paper in its actual form.

Reviewer #2

(Remarks to the Author)

In the present work, a third stage of warm rolling was introduced to develop high strength and high ductility combined RAFM steel, before quenching and tempering. Obviously, the strength was slightly improved by about 40 MPa, and fracture elongation was significantly improved to 49% (an increase of ~18%). This is an uncommon progress in high performance RAFM steel. The fundamental reason for the improvement was attributed to the unique trimodal multiscale microstructure comprising nanoscale and microscale ferrite, and tempered martensite with low-angle nanograins by authors. But I think there is lack of sufficient evidence to support this conclusion. I have the following doubts about it.

Major points:

1. The main novelty of this article claimed by authors is the development of trimodal multiscale microstructure. Unfortunately, there is no obvious evidence to show the trimodal microstructure. The main evidence is shown by Fig. 2 and Fig. S4. However, it looks very similar for the so-called bimodal and trimodal microstructure from Fig. 2a and 2e. The only difference that I can find is the different thickness of the lines for boundaries. Moreover, there is no obvious peak can be detected in Fig. S4. And by comparing Fig. 2a and 2e, it looks like the fraction of the large ferrite grains in bimodal sample is higher than trimodal sample. So, I strongly doubt the authenticity of the existence of the trimodal structure. Please check carefully and present them more clearly.

2. The mechanism of the formation of the trimodal microstructure is unclear. Although there is a schematic explanation in Fig. 1. There is no evidence to show the microstructure evolution during the processing with and without stage 3. Therefore, the schematic illustration in Fig.1 is lack of support. For example, how did the nanograin form? Why did you say the nanograin came from the ferrite transformation from small austenite grains during quenching? Why did the small austenite

grains transform to martensite during quenching? Nanograins can also be formed during aging by recrystallization of martensite at a such high aging temperature, this is more reasonable. If so, how did the stage 3 affect the formation of microstructure. All of these needs clear evidence to show microstructure evolution and to clarify the effect of stage 3 on microstructure development.

3. The fundamental reason for the improved strength was recognized as precipitation strengthening by nanosized MX precipitates. This is undoubtable. But the fundamental underlying for improved ductility, which is the key novelty of this work, was not clear in current state. Authors claimed the trimodal microstructure may play a key role in the enhancement of tensile ductility. This statement was based on the observation from Fig. 4 and Fig.5. Especially, it was concluded that nanograins in trimodal microstructure from martensite was free of strain (Fig. 4p and q) and promoted formation of stable fine voids (Fig.5), hence these features are likely to be a feature of high tensile ductility and not ductility limiting. This statement is highly speculative and lacks understanding of the physical mechanisms of metals. In my opinion, the nano-sized precipitates may play a significant role in the high ductility. As authors observed in Fig 4d, i, n and s, during tensile, slip bands were likely formed in fine grains at the early and middle of tensile deformation, and network cell structure was formed till strain to 38%, then fracture till to strain of 49%. This indicated that the nanograins was of sufficient ability to multiply dislocations after cell structure formation in the trimodal sample, leading to a high later deformation capability to 49%, whereas, those grains in the bimodal sample were of insufficient ability to multiply dislocations after cell formation, leading to less later deformation capability (since there is no comparable study of the bimodal sample, this needs to be check in details. In addition, the high ability of dislocation multiplication in cells may be attributed to the high density of nanosized precipitates in the trimodal sample. Therefore, I suggest to discuss clearly on this aspect. Therefore, I recommend to reject it.

Minor points:

1. It is suggested to study the precipitates by SANS for the bimodal sample, rather than Eurofer97. This will provide strong evidence to show the effect of stage 3 on precipitates.
2. The observation of dislocations under transmission electron microscopy has directionality. Generally, dislocation states should be observed under the same two-beam condition. How about in this work, like Fig. 2d and 2h.
3. The phrase of normalisation should be revised to be re-austenization, as quenching was carried out after holding at 980C. For normalisation, air cooling is usually performed.
4. The slight difference in Cr content in Fig 2j can not verify the partitioning of Cr from large gamma grains to fine grains.
5. The characterization of Cr₂₃C₆ and (Ti,V)C is poor, diffraction pattern is suggested to be carried out for identification of crystal structures for precipitates.

Reviewer #3

(Remarks to the Author)

This study investigates the enhancement of both strength and ductility in reduced activation ferritic/martensitic (RAFM) steels through the implementation of a three-stage thermomechanical manufacturing process, resulting in a trimodal multiscale microstructure. The achieved mechanical properties surpass those of conventional Eurofer97 steel. However, the novelty of this research may not meet the criteria for publication in Nature Communications, and the manuscript organization requires improvement. The main comments are as follows:

(1) Mechanical properties: The comparison of yield strength and total elongation with literature data lacks consideration of the influence of specimen dimensions. The gauge length of ~12.5 mm is smaller than the standard sample, which could overstate the ductility. Thus employing uniform elongation for comparison is recommended.

(2) Microstructure analysis: The microstructure of the RAFM steel needs to be further analyzed.

-the microstructure of the RAFM steel was not well exhibited using the EBSD map in Fig. 2a and e, Additional SEM morphologies are required to show the ferrite, martensite, and carbides more clearly.

-The authors highlight the trimodal structure of the RAFM steel. However, the trimodal structure mentioned should be reflected in grain size distribution, which currently does not exhibit the expected nano and micron grains.

-the authors state that the carbides have significant influences on the microstructure formation and mechanical properties of the RAFM steel. However, the carbides are roughly analyzed using EDS mapping. Further EDS point scanning and high-resolution TEM analysis should be performed to determine the structure of the carbides

-the initial dislocation density also contributes to the yield strength. Therefore, the dislocation density in different samples needs to be quantified.

(3) strain hardening analysis: the authors conducted the interrupted tensile tests to investigate the work hardening behavior. The interrupted strains are selected based on the true stress strain and strain hardening rate curves in Fig. S6. Drawing true stress-strain and strain hardening rate curves with the same vertical coordinates would facilitate the identification of the point of uniform elongation. In addition, it is suggested to focus on the microstructural evolution in the homogeneous deformation stage to study the strain hardening mechanisms, rather than the post uniform elongation stage.

(4) The current manuscript lacks profound discussion on the underlying mechanisms contributing to the enhanced mechanical properties of RAFM steel. The enhanced tensile strength should be quantitatively analyzed from the contribution of the trimodal structure and carbide. It seems that the contribution of tempered martensite to the mechanical properties is ignored in current manuscript. The refined martensite grains also benefit the strength and elongation of the RAFM steel.

(5) The fine new strain free grains formed during tensile deformation is not well supported by the experiments. Identification of new formed ferritic grains (<100 nm) using EBSD may be challenging with the current step size. of 0.15-0.1 μm. How do the authors determine the grains in Fig. 4r are the new formed ferritic grains rather than the initial nano grains? Furthermore, the formation mechanisms of the described new formed grains are not involved in this manuscript.

Version 1:

Reviewer comments:

Reviewer #1

(Remarks to the Author)

After going through the author's response to my comments, I believe they addressed them in a proper manner and I am satisfied with the revision performed. I believe the paper is now in a better shape and address the problem tackled in a more comprehensive manner to the reader. I recommend therefore the acceptance of the paper in its present form. Best wishes,

Reviewer #2

(Remarks to the Author)

There are two main problems remained, as below:

1. The presented microstructure evolution does not give me a clear image to show how different between samples with and without stage 3. Systematical studies by OM, SEM, should be comparatively carried out to show the difference in microstructure evolution after each processing stage. rather than only by drawing a schematic diagram, like Fig.1. These results could be provided in supplementary materials to support Fig.1.
2. In the file of "Response to referees", there is a statement like this: "We did say that smaller austenite grains may transform to ferrite (see point above); this is the well-known factor that austenite grain size controls Ms temperature, and as the grain size becomes too small, martensite formation maybe inhibited." I think this is totally wrong. The cited reference shown that austenite grain refinement leads to a decrease in the MS temperature. But even MS temperature was decreased by refined austenite grain size, they should transform to martensite or remained to be retained as austenite, rather than transform to ferrite. Ferrite transformation should be taken place at high temperature region, it is depended on chemical composition and cooling rate.

Reviewer #3

(Remarks to the Author)

The authors have addressed part of the raised comments. However, I think the current manuscript does not meet the requirement of publication in Nature Communications. The following comments should be considered:

- (1) It is correct that a range of specimen sizes are used for the tensile data are reported in the literature. Thus, the comparison of total elongation with literatures is meaningless. It is suggested to remove Fig. 1c. Figure 1b is enough to verify the better total elongation of the stage 3 steel than the Eurofer97.
- (2) I think one of the novelties in this manuscript is the occurrence of the newly formed dislocation free ferrite grains during deformation. The authors deemed that they play an important role in delaying microvoid coalescence and the resulting higher post necking deformation. It is necessary to clarify the formation mechanisms of these dislocation free ferrite grains from the microstructural aspects. Do these free ferrite grains forms in the stage 2 or Eurofer97 steels after deformation? The corresponding deformed microstructure should be provided.
- (3) In the conventional DP steel, the large strength mismatch between ferrite and martensite produces a high strain gradient near phase boundaries, resulting in the dissociation of boundaries and formation of high-density voids. In the current manuscript the aging treatment decrease the strength of martensite. The lower deformation incompatibility between tempered martensite and ferrite improves co-deformation of two phases. This could decrease the number of voids at boundaries and improve the elongation. Thus, the role of the dislocation free grains in tensile ductility should be reconsidered.

Version 2:

Reviewer comments:

Reviewer #2

(Remarks to the Author)

Most of revisions meet me well. There are three points need to be made more clear.

1. As authors said on page 5, "On quenching to room temperature, the larger γ_5 grains transform to martensite (α'_2). Microstructural inspection indicates that the smaller γ_3 and γ_4 grains transform to ferrite (α_3 and α_4 respectively) on cooling." I think it need to give the microstructure after quenching from normalisation at 980 degree C. It can make readers to understand better on ferrite transformation from smaller austenite grain size by quenching. In SEM image, it is easy to identify quenched martensite and ferrite.
2. The rolling parameters like strain rate, passes and reduction of each pass at the three stages should be given.
3. The details on the method used for dislocation density measurements should be given. In addition, table 1 listed the dislocation strengthening value, rather than dislocation density.

Reviewer #3

(Remarks to the Author)

The authors have addressed my comments

Gong et al. NCOMMS-24-09362, Response to referees

Reviewer #1 (Remarks to the Author):

The novelty of this work consists on proposing a novel thermomechanical processing for RAFM steels that enhances simultaneously tensile strength and ductility by a considerable refining of the final microstructure formed. The key-factor for such behavior is the combination of warm-rolling stage where DIFT mechanism is activated, and a subsequent normalization treatment to generate a final ultrafine martensite grain microstructure. This novel processing, which has been studied profusely for steel processing in automotive sector applied steels, has not been reported previously for RAFM steels. Therefore, this paper reports a work of significance for the field.

Regarding literature review of the state-of-the-art, the references indicated are fairly modern with well selected number of seminal papers on the matter. I miss some of earlier works on this subject such as:

N. Baluc et al, Journal of Nuclear Materials, 367-370 (2007) 33-41
H. Tamigawa et al, Fusion Engineering & Design, 83 (2008) 1471-1478
The work reported in this paper support the conclusion raised and is methodology sound, providing enough details to reproduce the tests.

Response: Thank you. These papers have been added.

I could not find any flaws in the paper. My only concern is all the tests performed are focused on improving strength-ductility compromise, but not dynamic mechanical testing were performed in order to check if this claimed improvement is really achieved. No creep tests on irradiated samples have been performed.

Response: Indeed, dynamic and creep testing is important. We have tests ongoing and will publish these in due course.

Having all the above mentioned reasons in mind, I recommend the acceptance of the paper in tis actual form.

Reviewer #2 (Remarks to the Author):

In the present work, a third stage of warm rolling was introduced to develop high strength

and high ductility combined RAFM steel, before quenching and tempering. Obviously, the strength was slightly improved by about 40 MPa, and fracture elongation was significantly improved to 49% (an increase of ~18%). This is an uncommon progress in high performance RAFM steel. The fundamental reason for the improvement was attributed to the unique trimodal multiscale microstructure comprising nanoscale and microscale ferrite, and tempered martensite with low-angle nanograins by authors. But I think there is lack of sufficient evidence to support this conclusion. I have the following doubts about it.

Major points:

1. The main novelty of this article claimed by authors is the development of trimodal multiscale microstructure. Unfortunately, there is no obvious evidence to show the trimodal microstructure. The main evidence is shown by Fig. 2 and Fig. S4. However, it looks very similar for the so-called bimodal and trimodal microstructure from Fig. 2a and 2e. The only difference that I can find is the different thickness of the lines for boundaries. Moreover, there is no obvious peak can be detected in Fig. S4. And by comparing Fig. 2a and 2e, it looks like the fraction of the large ferrite grains in bimodal sample is higher than trimodal sample. So, I strongly doubt the authenticity of the existence of the trimodal structure. Please check carefully and present them more clearly.

Response: We have adjusted the text in a major way to make the differences clearer. The data presented in Fig. S4 is derived from EBSD and therefore only shows the grain size distribution grains with a size $> \sim 1 \mu\text{m}$. As shown in Fig. 2, the major differences are in the grains $< \sim 1 \mu\text{m}$, which are only adequately imaged in the TEM and deriving a size distribution from TEM images in such a heterogeneous microstructure is very difficult.

2. The mechanism of the formation of the trimodal microstructure is unclear. Although there is a schematic explanation in Fig. 1. There is no evidence to show the microstructure evolution during the processing with and without stage 3. Therefore, the schematic illustration in Fig.1 is lack of support. For example, how did the nanograin form? Why did you say the nanograin came from the ferrite transformation from small austenite grains during quenching? Why did the small austenite grains transform to martensite during quenching? Nanograins can also be formed during aging by recrystallization of martensite at a such high aging temperature, this is more reasonable. If so, how did the stage 3 affect the formation of microstructure. All of these needs clear evidence to show microstructure evolution and to clarify the effect of stage 3 on microstructure development.

Response: We are not sure we understand these comments. The text under the heading "Processing" is entirely devoted to how the microstructure evolves at each stage. We cannot see how this could be clearer.

The referee asks us "to show the microstructure evolution during the processing with and without stage 3." But the stage 2 microstructure is with and without stage 3 and is described

in detail. The evolution of nanograins is described in detail, clearly describing the origin of the nanograins.

The referee states: “Nanograins can also be formed during aging by recrystallization of martensite at a such high aging temperature, this is more reasonable.” While this is possible, we did not see it. The substructure within the martensite is easy to identify since the lath boundaries are clearly visible (see for example in Fig. 2c and g) and the substructure within the lath boundaries comprised of low angle boundaries, as clear stated in the text. Moreover, recrystallisation within martensite results in dislocation free grains which consume the lath boundaries, as shown by: <https://doi.org/10.2355/isiinternational.41.1047>

The referee states: “Why did the small austenite grains transform to martensite during quenching?” We do not state that. We said: “On quenching to room temperature, the larger γ_5 grains transform to martensite (α'_2) and the smaller γ_3 and γ_4 grains transform to ferrite (α_3 and α_4 respectively).” It is frequently observed experimentally that austenite grain refinement leads to a decrease in the MS temperature, which the smaller size of austenite grain, the lower the Ms temperature with prefer to transfer to ferrite.

Ref []:<https://doi.org/10.2355/isiinternational.53.2218>

The referee states: “Why did you say the nanograin came from the ferrite transformation from small austenite grains during quenching?” We cannot find anywhere we made that statement. What we stated was “In Fig. 2(a, e), the short chains of α_4 necklace grains are observed using Electron Backscattered Diffraction (EBSD). These can be distinguished from the tempered martensite grains as the latter have an abundance of Low Angle Grain Boundaries (LAGBs, red lines) within each grain (high angle grain boundaries are in black). “We did say that smaller austenite grains may transform to ferrite (see point above); this is the well-known factor that austenite grain size controls Ms temperature, and as the grain size becomes too small, martensite formation maybe inhibited. The reality is that one cannot tell whether the finer austenite transformed to martensite or ferrite. We believe this is clear in the text.

We have checked the text again and made minor modifications to make these points clearer.

3. The fundamental reason for the improved strength was recognized as precipitation strengthening by nanosized MX precipitates. This is undoubtable. But the fundamental underlying for improved ductility, which is the key novelty of this work, was not clear in current state. Authors claimed the trimodal microstructure may play a key role in the enhancement of tensile ductility. This statement was based on the observation from Fig. 4 and Fig.5. Especially, it was concluded that nanograins in trimodal microstructure from martensite was free of strain (Fig. 4p and q) and promoted formation of stable fine voids (Fig.5), hence these features are likely to be a feature of high tensile ductility and not ductility limiting. This statement is highly speculative and lacks understanding of the physical mechanisms of metals. In my opinion, the nano-sized precipitates may play a significant role

in the high ductility. As authors observed in Fig 4d, i, n and s, during tensile, slip bands were likely formed in fine grains at the early and middle of tensile deformation, and network cell structure was formed till strain to 38%, then fracture till to strain of 49%. This indicated that the nanograins was of sufficient ability to multiply dislocations after cell structure formation in the trimodal sample, leading to a high later deformation capability to 49%, whereas, those grains in the bimodal sample were of insufficient ability to multiply dislocations after cell formation, leading to less later deformation capability (since there is no comparable study of the bimodal sample, this needs to be check in details. In addition, the high ability of dislocation multiplication in cells may be attributed to the high density of nanosized precipitates in the trimodal sample. Therefore, I suggest to discuss clearly on this aspect.

Response: Thank you for these comments and we accept that this important point was not fully developed. We have made major changes to the discussion on this point.

Minor points:

1. It is suggested to study the precipitates by SANS for the bimodal sample, rather than Eurofer97. This will provide strong evidence to show the effect of stage 3 on precipitates.

Response: It would indeed be good to undertake SANS of the bimodal sample. However, waiting times for such experiments are long and would unacceptably delay publication of this paper.

The purpose of this paper is to develop a new manufacturing process to produce microstructures with a balanced combination of strength and ductility, compared to conventional Eurofer 97 steel. Our research also includes the design of bimodal microstructures as part of our development process, although these are not the final microstructures we have designed.

2. The observation of dislocations under transmission electron microscopy has directionality. Generally, dislocation states should be observed under the same two-beam condition. How about in this work, like Fig. 2d and 2h.

Response: Indeed, dislocation imaging should always be undertaken under well-defined 2-beam conditions. In our case we used STEM not TEM to image dislocations and so the exact diffraction condition is not as important. We have added a statement to this effect.

3. The phrase of normalisation should be revised to be re-austenization, as quenching was carried out after holding at 980C. For normalisation, air cooling is usually performed.

Response: This statement is true, but throughout the RAFM literature (e.g. Xiang Chen, Arunodaya Bhattacharya, Mikhail A. Sokolov, Logan N. Clowers, Yukinori Yamamoto, Tim Graening, Kory D. Linton, Yutai Katoh, Michael Rieth, Mechanical properties and microstructure characterization of Eurofer97 steel variants in EUROfusion program, Fusion

Engineering and Design, Volume 146, Part B, 2019, pp 2227-2232), the term “normalisation” is used with a quench. Nevertheless, we have changed this.

4. The slight difference in Cr content in Fig 2j cannot verify the partitioning of Cr from large gamma grains to fine grains.

Response: We accept that, while partitioning is probable, we do not have adequate evidence for this and have adjusted the wording accordingly.

5. The characterization of Cr₂₃C₆ and (Ti,V)C is poor, diffraction pattern is suggested to be carried out for identification of crystal structures for precipitates.

Response: Of course, this was carried out. The results have been added to the supplementary material.

Cr₂₃C₆

[011]
FCC, a=b=c=1.062 nm

VC

Reviewer #3 (Remarks to the Author):

This study investigates the enhancement of both strength and ductility in reduced activation ferritic/martensitic (RAFM) steels through the implementation of a three-stage thermomechanical manufacturing process, resulting in a trimodal multiscale microstructure. The achieved mechanical properties surpass those of conventional Eurofer97 steel. However, the novelty of this research may not meet the criteria for publication in Nature Communications, and the manuscript organization requires improvement. The main comments are as follows:

(1) Mechanical properties: The comparison of yield strength and total elongation with literature data lacks consideration of the influence of specimen dimensions. The gauge length of ~12.5 mm is smaller than the standard sample, which could overstate the ductility. Thus employing uniform elongation for comparison is recommended.

Response: Yes, this is correct. We have included the uniform elongation, Table S3, and made this point in the discussion. It is nevertheless important to note that the post necking deformation was considerably higher for the stage 3 material compared to the Eurofer97. We note that a range of specimen sizes are used for the tensile data are reported in the literature, which makes absolute comparisons difficult across all published work.

(2) Microstructure analysis: The microstructure of the RAFM steel needs to be further analyzed.

-the microstructure of the RAFM steel was not well exhibited using the EBSD map in Fig. 2a

and e, Additional SEM morphologies are required to show the ferrite, martensite, and carbides more clearly.

Response: We are not clear how we should further analyse the microstructure. We have presented SEM, EBSD and TEM/STEM images of the structures. Actually, SEM images do not represent the morphologies very well as it difficult to differentiate between the ferrite and tempered martensite in such images. Moreover, the entire fraction of MC type carbides cannot be seen in SEM images, being typically only 6nm. Examples of SEM images are shown below:

-The authors highlight the trimodal structure of the RAFM steel. However, the trimodal structure mentioned should be reflected in grain size distribution, which currently does not exhibit the expected nano and micron grains.

Response: The data presented in Fig. S4 is derived from EBSD and therefore only shows the grain size distribution grains with a size $> \sim 1 \mu\text{m}$. As shown in Fig. 2, the major differences are in the grains $< \sim 1 \mu\text{m}$, which are only adequately imaged in the TEM and deriving a size distribution from TEM images in such a heterogeneous microstructure is very difficult.

-the authors state that the carbides have significant influences on the microstructure formation and mechanical properties of the RAFM steel. However, the carbides are roughly analyzed using EDS mapping. Further EDS point scanning and high-resolution TEM analysis should be performed to determine the structure of the carbides.

Response: Thank you for pointing out this deficiency. This is now included in the additional material. The precipitate size and fractions are now quantified and added to the text (detailed further below).

-the initial dislocation density also contributes to the yield strength. Therefore, the dislocation density in different samples needs to be quantified.

Response: Thank you for pointing out this deficiency. The dislocation density has been measured and added to the manuscript and included in the prediction of yield strength (detailed below).

(3) strain hardening analysis: the authors conducted the interrupted tensile tests to investigate the work hardening behavior. The interrupted strains are selected based on the true stress strain and strain hardening rate curves in Fig. S6. Drawing true stress-strain and strain hardening rate curves with the same vertical coordinates would facilitate the identification of the point of uniform elongation. In addition, it is suggested to focus on the microstructural evolution in the homogeneous deformation stage to study the strain hardening mechanisms, rather than the post uniform elongation stage.

Response: We are sorry, but do not understand this comment. Both true stress/true strain and work hardening rate are drawn on the same graph in Fig. S6 (now S7). To draw them with the same vertical coordinates would lead to a massive y axis given that the range of true stress is from 0-800MPa and the work hardening rate is from -4000,000 to + 4000,000 MPa. We have identified the point of uniform elongation and added it to the text. We did indeed include an analysis of the deformation during uniform elongation, which is, as the referee indicates, the most important region.

It is important to point out that since we have substantially better post uniform elongation compared to the standard Eurofer steel it is important to explain the origin of this. During this study we observed the newly formed dislocation free ferrite grains, which clearly play a part in delaying microvoid coalescence.

(4) The current manuscript lacks profound discussion on the underlying mechanisms contributing to the enhanced mechanical properties of RAFM steel. The enhanced tensile strength should be quantitatively analyzed from the contribution of the trimodal structure and carbide. It seems that the contribution of tempered martensite to the mechanical

properties is ignored in current manuscript. The refined martensite grains also benefit the strength and elongation of the RAFM steel.

Response: Thank you for the comment and we acknowledge this deficiency in the original manuscript. The relative contributions from each microstructural component have been calculated and added to the manuscript. As part of this, we have measured additional values such as dislocation density.

(5) The fine new strain free grains formed during tensile deformation is not well supported by the experiments. Identification of new formed ferritic grains (<100 nm) using EBSD may be challenging with the current step size of 0.15-0.1 μm . How do the authors determine the grains in Fig. 4r are the new formed ferritic grains rather than the initial nano grains? Furthermore, the formation mechanisms of the described new formed grains are not involved in this manuscript.

Response: The grains were determined from both TEM images and TKD-EBSD (transmission-electron backscatter diffraction); both had adequate resolution to image such features. In both cases, the grains are strain free, see for example Figs. 4(k) and 4(p). These were always associated with a high angle boundary. No dislocation-free grains were found in the starting material. Furthermore, it is very strange to find dislocation-free grains in the strained material. We have addressed formation in the text and have updated the text to make it clearer. However, it is very difficult to find a robust explanation.

Reviewer #1 (Remarks to the Author):

After going through the author's response to my comments, I believe they addressed them in a proper manner and I am satisfied with the revision performed. I believe the paper is now in a better shape and address the problem tackled in a a more comprehensive manner to the reader. I recommend therefore the acceptance of the paper in its present form.
Best wishes,

Reviewer #2 (Remarks to the Author):

There are two main problems remained, as below:

1. The presented microstructure evolution does not give me a clear image to show how different between samples with and without stage 3. Systematical studies by OM, SEM, should be comparatively carried out to should the difference in microstructure evolution after each processing stage. rather than only by drawing a schematic diagram, like Fig.1. These results could be provided in supplementary materials to support Fig.1.

Response: We politely note that SEM images of with and without stage 3 are given in Fig. 2. Optical microscopy does not have sufficient resolution to show the microstructures clearly, as illustrated with the OM image below. Conventional SEM also fails to reveal the microstructural differences clearly, as shown below. However, we have now added further images to the supplementary material to clarify the difference in stage 2 and stage 3.

2. In the file of "Response to referees", there is a statement like this: We did say that smaller austenite grains may transform to ferrite (see point above); this is the well-known factor that austenite grain size controls Ms temperature, and as the grain size becomes too small, martensite formation maybe inhibited." I think this is totally wrong. The cited reference shown that austenite grain refinement leads to a decrease in the MS temperature. But even MS temperature was decreased by refined austenite grain size, they should transform to martensite or remained to be retained asustenite, ranther than transform to ferrite. Ferrite transformation should be taken place at high temperature region, it is depended on chemical composition and cooling rate.

Response: Thank you for this comment. We accept that the referee is entirely correct. We have made significant alterations to the text in this respect, marked in red.

Reviewer #3 (Remarks to the Author):

The authors have addressed part of the raised comments. However, I think the current manuscript does not meet the requirement of publishment in Nature Communications. The following comments should be considered:

(1) It is correct that a range of specimen sizes are used for the tensile data are reported in the literature. Thus, the comparison of total elongation with literatures is meaningless. It is suggested to remove Fig. 1c. Figure 1b is enough to verify the better total elongation of the stage 3 steel than the Eurofer97.

Response: We accept this point and Fig. 1c has been removed.

(2) I think one of the novelties in this manuscript is the occurrence of the newly formed dislocation free ferrite grains during deformation. The authors deemed that they play an important role in delaying microvoid coalescence and the resulting higher post necking deformation. It is necessary to clarify the formation mechanisms of these dislocation free ferrite grains from the microstructural aspects. Do these free ferrite grains forms in the stage 2 or Eurofer97 steels after deformation? The corresponding deformed microstructure should be provided.

Response: This is a very interesting question, one which we have thought about at length. No, we do not see these features in stage 2 material, but they are only seen at the high strains (35%) observed in the stage 3 steel (the stage 2 steel had failed by this strain). Hence, we cannot add images that directly compare stage 2 and 3 at this strain. We have added text to make this clear.

We have now added text to discuss the origin of these strain free grains.

(3) In the conventional DP steel, the large strength mismatch between ferrite and martensite produces a high strain gradient near phase boundaries, resulting in the dissociation of boundaries and formation of high-density voids. In the current manuscript the aging treatment decrease the strength of martensite. The lower deformation incompatibility between tempered martensite and ferrite improves co-

deformation of two phases. This could decrease the number of voids at boundaries and improve the elongation. Thus, the role of the dislocation free grains in tensile ductility should be reconsidered.

Response: This is an interesting point that the referee makes, that we did not fully consider in the original submission. We agree with this argument and have added text to address this point.

We thank the referee for the further comments. All comments have been addressed in full.

Reviewer #2 (Remarks to the Author):

Most of revisions meet me well. There are three points need to be made more clear.

1. As authors said on page 5, "On quenching to room temperature, the larger γ_5 grains transform to martensite (α'_2). Microstructural inspection indicates that the smaller γ_3 and γ_4 grains transform to ferrite (α_3 and α_4 respectively) on cooling." I think it need to give the microstructure after quenching from normalisation at 980 degree C. It can make readers to understand better on ferrite transformation from samller austenite grain size by quenching. In SEM image, it is easy to identify quenched martensite and ferrite.

Response: We have added a figure to the supplementary procedure to show this and adjusted the text accordingly.

2. The rolling parameters like strain rate, passes and reduction of each pass at the three stages should be given.

Response: The rolling parameters have been added in Table S2.

3. The details on the method used for dislocation density measurements should be given. In addition, table 1 listed the dislocation strengthening value, rather than dislocation density.

Response: While the measurement of dislocation density is quite standard, we have now added the procedure to the methods section. The measured dislocation density was already given in Table S4 (now Table S5) along with all the other input values.